Distribution and abundance of the land snail Pollicaria elephas (Gastropoda: Pupinidae) in limestone habitats in Perak, Malaysia

Liew Thor-Seng thorseng@ums.edu.my 1
Phung Chee-Chean 1
Mat Said Mohamad Afandi 2
Hoo Pui Kiat 1 3
1 Institute for Tropical Biology and Conservation, Universiti Malaysia Sabah , Kota Kinabalu , Malaysia
2 Associated Pan Malaysia Cement , Chemor , Perak , Malaysia
3 Faculty of Science, University of Malaya , Kuala Lumpur , Malaysia
Oehlmann Jörg
Electronic publication date: 2021 Jul 28
Publication date: 2021
Volume: 9
Electronic Location ID: e11886
Received 2020 Sep 28; Accepted 2021 Jul 9
Copyright: ©2021 Liew et al.
Copyright year: 2021
Copyright holder: Liew et al.
License: This is an open access article distributed under the terms of the Creative Commons Attribution License, which permits unrestricted use, distribution, reproduction and adaptation in any medium and for any purpose provided that it is properly attributed. For attribution, the original author(s), title, publication source (PeerJ) and either DOI or URL of the article must be cited.
License URL: https://creativecommons.org/licenses/by/4.0/

Keywords: Elephant pupinid snails, Karst, Indochina, Malay Peninsula, Perak, Kinta Valley, Ecology

Funding: YTL Cement GLS0006 YTL Cement provided funding for this project (GLS0006), from 2017 to 2019, to Universiti Malaysia Sabah. The funders had no role in study design, data collection and analysis, decision to publish, or preparation of the manuscript.

==============================
This study aimed to reveal the habitat variables that determine the distribution and abundance of the land snail Pollicaria elephas in limestone habitats in Perak, Malaysia. Seventeen plots were selected on a limestone hill to determine the effect of environmental variables on the abundance of this land snail. The environmental variables we considered included habitat (canopy cover and leaf litter thickness), topography (elevation, aspect, ruggedness, and slope), microclimate (soil temperature, air temperature, and humidity), and vegetation (abundance of respective vascular plant species). The correlation analyses suggested that the snails’ abundance was positively correlated with the abundance of the four vascular plant species: Diospyros toposia var. toposoides, Croton cascarilloides, Kibatalia laurifolia, and Mallotus peltatus. Plots with lower soil temperatures had more snails than plots with higher soil temperatures. Our results show that plots in the southern part of the limestone hill, in which P. elephas were absent, were similar in habitat, topography, microclimate, and vegetation to the plots in the northern part of the limestone hill, where specimens were mostly present. The absence of this species in suitable habitats may be due to their low dispersal ability rather than adverse environmental conditions.

Introduction

There are approximately 1,000 recognised land snail species in Malaysia (http://malaypeninsularsnail.myspecies.info/, http://opisthostoma.myspecies.info/, http://borneanlandsnails.myspecies.info/). However, the ecology of the land snail species is often poorly known. To date, only a handful species from the genera Plectostoma, Georissa, Gyliotrachela, Diplommatina have been studied in terms of their growth (Berry, 1962; Berry, 1963; Liew et al., 2014b), reproduction (Berry, 1965), and demography (Berry, 1966; Schilthuizen et al., 2003).

Land snails of the genus Pollicaria, commonly known as elephant pupinid snails, belong to the family Pupinidae. All seven Pollicaria species and subspecies from Indochina and Peninsular Malaysia are endemic to this region. P. elephas is the only Pollicaria species found on Peninsular Malaysia (Kongim et al., 2013) (Fig. 1A). This species was described by De Morgan (1885) in the state of Perak, Malaysia. P. elephas specimens were recorded in various localities from the limestone hills in Perak and from two other locations in Pahang (Chan, 1998; Kongim et al., 2013; Foon, Clements & Liew, 2017; Minton, Harris & North, 2017; Global Biodiversity Information Facility, GBIF, 2018).

Figure 1 Photographs of specimens of Pollicaria elephas and the study site.

(A) Male individual of Pollicaria elephas. Scale: 10 mm. (B & C) Capture-mark-recapture experiment in the field. Different colours of nail polish were used to mark the shells of living Pollicaria elephas that were collected during different sampling sessions. (D & E) Habitat of Pollicaria elephas at two localities at northern part of the limestone hill.

Some aspects of P. elephas’ morphology, taxonomy, karyotypes, and geographical distribution have been studied (Pain, 1974; Chan, 1997; Kongim et al., 2010; Kongim et al., 2013). However, the ecology and small-scale distribution of this ground-dwelling species remain unknown. We found localities with very high densities of P. elephas during a preliminary survey of a limestone hill in Perak, while just tens of meters away, no snails were found. This patchy distribution is not unusual. A previous study on another similarly-sized land snail, Limicolaria martensiana, also showed an uneven distribution with more than 100 individuals per m2 at one locality at Uganda (Owen, 1969).

Studies of other macro snails from other regions show that a higher land snail abundance can be explained by vegetation or habitat characteristics, such as a denser and heterogeneous canopy and understory, higher litter humidity and thickness, older and bigger trees, rotten logs, and calcium availability (Boag, 1985; Martin & Sommer, 2004; Müller, Strätz & Hothorn, 2005; Horsák et al., 2007; Dvořáková & Horsák, 2012). The flora composition can be difficult to measure directly but may be a very useful predictor for snail communities (Dvořáková & Horsák, 2012).

We examined specific environmental parameters that may be responsible for the unevent distribution of P. elephas on a limestone hill. To date, the limestone hill in Perak is the only location where a sizeable living population of P. elephas is found based on several systematic samplings of land snails throughout limestone hills in Peninsular Malaysia (Chan, 1997; Clements et al., 2008; Foon, Clements & Liew, 2017). We first assessed the population size and density of P. elephas at different localities on the hill in Perak. We then examined the vegetation and topographic and microclimatic variables for each locality to characterise species-specific requirements.

Materials and Methods

Study site

The study site was located on a limestone hill in Perak, Malaysia. We established a total of 17 plots, each measuring 2 m × 4 m. Seven plots were located in the northern part, nine at the southern part, and one at the central part of the hill (Fig. 2). Each plot was located next to limestone rock outcrops. A pilot survey was conducted to ensure that these plots covered habitats with different environmental variables and to identify plots with living P. elephas suitable for the population density study. In each plot, leaf litter was searched manually by two people for over 20 min to find living snails and empty shells. Environmental variables for each of the 17 plots were measured during the pilot survey on 11 May 2018.

Figure 2 Study sites and location of the 17 studied plots at a limestone hill in the state of Perak, Malaysia.

(A) Topographic map showing the seven plots at the northern part, nine plots at the southern part and one plot (C-P4) at the central part of the hill. (B) Aerial photograph from the north-western flank (blue arrow) of the hill and location of the seven plots on a topographic map. Five of the plots were selected for the capture-mark-recapture study. The blue box indicates an area of 500 m × 150 m. (C) Aerial photograph from the southern flank (blue arrow) of the hill and the location of the nine plots on a topographic map.

Spatial distribution and population density

We did not find any living P. elephas or empty shells in the 11 of the 17 examined plots during the pilot survey (Tables S1). We used the capture-mark-recapture method (CMR) to study the population of P. elephas in five of the six plots with living snails. One of the plots (AP-0) was inaccessible after the pilot survey so the CMR study could not be conducted in that plot. The captured P. elephas were marked with different colours of nail polish (Figs. 1B and 1C) during different sampling sessions. The snails were then released back to their respective plots, and plots were resampled after 10 to 15 days. The CMR was conducted four times for all plots with the exception of one plot (AP-2), where only three CMR sessions took place. The first sampling was conducted on 9 July 2018 and the first recapture was on 19 July 2018, the second recapture occurred on 1 August 2018, and the third recapture took place on 16 August 2018. The collected snails from the following demographics were examined: juvenile (<3 whorls), subadult (3–5 whorls), and adult (with aperture lip). All living snails caught were released back to their respective plots while the empty shells were collected and deposited in the BORNEENSIS collection of Universiti Malaysia Sabah.

We calculated the P. elephas population density by counting the living snails in individual plots during the different CMR sessions. The population size of P. elephas was estimated based on the data collected using the Schnabel index:

• N = total number of snails (unknown)

• C = number of snails captured on the first sampling

• M = number of snails captured on subsequent sampling

• R = number of snails captured on both samplings

Multiple marks and recaptures ensured a more accurate estimation of the population size. The Schnabel method (Alcoy, 2013), which allows multiple capture-recapture encounters, was used:

N=∑i=1mMiCi ∑i=1mRi

Mi=Total number of previously marked snails at timei

Ci=The number caught at timei

Ri=The number of marked snails caught at timei

Environmental variables

We studied four main environmental variables, namely habitat, topography, microclimate, and vegetation. All of the variables were measured for all 17 plots, with the exception of the microclimate. We measured leaf litter thickness by averaging litter thickness at eight points within each plot for habitat variables and estimated the percentage of canopy cover. To obtain topographic variables, we created a digital elevation model (DEM) of 4 m2 cell-size based on a 5-meter interval contour map for each of the northern and the southern study site using Triangular interpolation (TIN) in QGIS (ver. 2.18.24; QGIS Development Team, 2018). We then used terrain analysis tools to derive topographic features including slope, aspect, and the terrain ruggedness index (a quantitative measurement of terrain heterogeneity) (Riley, DeGloria & Elliot, 1999). Topographic parameters were extracted for each sampling plot using the ‘Add raster values to point’ setting in SAGA (Conrad et al., 2015).

Microclimatic variables

We installed climatic HOBO data loggers to record air temperature and humidity (HOBO MX2301 Temperature/RH) approximately one meter above the ground for eight of the 17 plots (A-Polli3, A-Polli9, D-P1, D-P3, D-P8, D-P9, D-P11, D-P12). We recorded the soil temperature using HOBO MX2303 temperature sensors for the same eight plots. External sensors were fully covered by leaf litter. These eight plots have been chosen, as they represent localities with both absence and presence data and different population densities of these snails. Living specimens were found in two northern plots, namely, plot A-Polli9, which had a higher number of snails (15–16 individuals) and plot A-Polli3, which had a lower number of snails (1–2 individuals). There were no living specimens found in the five southern plots. The climatic parameters were logged every 10 min in July 2018. Data from the two soil temperature loggers could not be retrieved due to damage by rain and wildlife.

Vegetation data

We counted and identified all the vascular plants with diameter at breast height (DBH) above one cm within a 5-meter radius from the centre of each of the 17 plots to obtain the number of vascular plant individuals and the number of vascular plant species of each plot. Voucher specimens were collected for each species and were subsequently identified by P.K. Hoo based on the reference materials at the Herbarium of Forest Research Institute Malaysia (FRIM).

Data analysis

Principal component analysis (PCA) was conducted to assess the degree of habitat heterogeneity among the 17 plots based on the two habitat variables (leaf litter thickness, and canopy cover), four topographic variables (elevation, slope, aspect, and ruggedness index) and two vegetation variables (number of vascular plant individuals and number of vascular plant species). The abundance of vascular plant species was not included in the PCA analysis due to the number of missing values in the dataset and the absence of certain plant species in plots. We visually explored the PCA plot for habitat heterogeneity according to the plots’ locations on the limestone hill (northern part, southern part, and central part). The analysis was done using R (File S2).

Correction tests were performed to examine any significant relationships between the abundance of snails and each of the habitat, vegetation, and topographic variables. We excluded 43 vascular plant species that were recorded only in one plot before statistical analysis to obtain vegetation data. The final dataset consisted of the abundance data for 20 vascular plant species from 17 plots.

As the data were not normally distributed, we used Spearman correlation testing based on both null-hypothesis significance testing and corroborated our analysis by using the Bayes factor (BF10) (Kass & Raftery, 1995). Our conclusion is based on the inference of both frequentist (p values) and Bayesian (Bayes factor) analyses. All analyses were performed using JASP software version 0.12.2 (JASP Team, 2020; File S3).

There was either complete or partial missing microclimatic data from July for some of the plots, so we did not calculate the mean values for each of the microclimatic variables per month. Hence, we could not perform rigorous statistical analysis to test the relationship between the microclimatic variables and the abundance of snails in the plot. Nevertheless, we explored the relationships between the abundance of snails and microclimatic variables by plotting the mean of each sampled plot’s daily microclimatic variables patterns. We calculated the daily mean air humidity, minimum air humidity, mean air temperature, maximum air temperature, mean soil temperature, and maximum soil temperature.

Results

Spatial distribution and population density of the snails

The numbers of living specimens collected over the four CMR sessions are shown in File S1). Living specimens were found in six out of seven plots on the northern part of the hill; none were found in plots on the central and southern parts (File S1). The smallest marked specimen was nine mm (shell width), and the majority of the marked snails were subadult and adult (File S1). The recapture rates were greater than 80% for the three plots with more than ten snails recorded during the pilot survey and the first capture session of CMR (A-Polli2, A-Polli7, and A-Polli9, see Tables S1–S4 in, except for two recapture sessions in plot A-Polli2 (23% and 67%). The recapture rates were between 50% and 100% for the two plots with less than ten snails.

Of the five plots examined in the CMR study, plot A-Polli2 had the highest population size of Pollicaria elephas (Table 1), and the calculated population density was estimated to be approximately 57 individuals for that plot and its surrounding area. The highest number of snails recorded per sampling event per plot was 26 specimens in plot A-Polli2 (Table S2). The snails’ population density in the sampling plots varied only slightly during the different sampling sessions for each plot (File S1).

Table 1 Population estimation for Pollicaria elephas for each plot based on capture-markrecapture technique and Schnabel index.

Plots	Population densitya	Population estimation by CMR (number of snails)	
A-Polli2	23.0 ± 3.4	56.9	
A-Polli9	15.8 ± 1.6	20.8	
A-Polli7	17.4 ± 2.8	19.5	
A-Polli3	1.4 ± 0.5	2.0	
A-P2	6.3 ± 2.1	12.1	
Notes.

a Average number of snails ± standard deviation for all of the CMR sessions at each 8 m2 plot.

Effect of environmental variables on snail occurrence and abundance

The first three PCA axes explained 78.7% of the habitat, topography, and vegetation variations between plots (Fig. 3, Files. S4, S5). As shown in Fig. 3, the PCA plot did not show apparent differences between the plots in the northern part (most of the plots with living P. elephas) and the southern part (all plots without living P. elephas) on the limestone hill. The abundance of P. elephas per plot was not correlated with canopy cover, leaf litter thickness, elevation, aspect, slope, and the ruggedness of the habitat (Table 2).

Figure 3 Principal components analysis (PCA) plot of the first two axes for habitat, topography, and vegetation variables for the 17 plots.

The colour of the labels represents the plot location on the limestone hill and red vectors represent the habitat, topography, and vegetation variables. No individuals of Pollicaria elephas were found in plots at the southern and central part of the limestone hill, while at the northern part living snails were found in six of the seven plots (none were found in A-P14).

The plot A-Polli9 with the higher number of living snails (15–16 individuals) had a soil mean temperature lower than 25 °C, and maximum temperature lower than 26 °C (Fig. 4). However, the two plots, namely D-P1 and D-P11, at the southern part of the hill with no snails recorded had a similar mean and maximum soil temperature (∼25 °C) to the plot A-Polli9.

Table 2 Correlation between the abundance of Pollicaria elephas for each plot with the habitat and topographic parameters.

Habitat and topographic variables	Bayesian Kendall’s Tau Correlations	BF10	Kendall’s Tau Correlations	p-value	
Number of vascular plant individuals for each plot	−0.186	0.512	−0.186	0.353	
Number of vascular plant species for each plot	0.128	0.392	0.128	0.525	
Canopy cover (%)	0.261	0.84	0.261	0.216	
Leaf litter thickness (cm)	0.154	0.437	0.154	0.436	
Elevation (meters)	0.183	0.504	0.183	0.355	
Aspect (counter-clockwise in degrees from 0 (due north) to 360 (again due north))	0.086	0.343	0.086	0.662	
Ruggedness index	0.092	0.372	0.092	0.669	
Slope (degrees)	0.241	0.721	0.241	0.224	

Figure 4 Soil temperature for each day in July 2018 in the six plots.

A-Polli3 and A-Polli9 are the plots with living Pollicaria elephas land snails, while the other four plots are without living P. elephas land snails. The insets represent the same plots with a different colour legend for the plots with living snails. (A) Mean soil temperature for each day. (B) Maximum soil temperature for each day.

Figure 5 Air temperature for each day in July 2018 in the eight plots.

A-Polli3 and A-Polli9 are the plots with living Pollicaria elephas land snails, while the other six plots are without living P. elephas land snails. The insets represent the same plots with a different colour legend for the plots with living snails. (A) Mean air temperature for each day. (B) Maximum air temperature for each day.

Figure 6 Humidity for each day in July 2018 in the seven plots.

Plot A-Polli3 is the plot with living Pollicaria elephas land snails, while the other six plots are without living P. elephas land snails. The insets represent the same plots with a different colour legend for the plots with living snails. (A) Mean humidity for each day. (B) Maximum humidity for each day.

All the plots at the southern and northern parts of the hill had a similar mean temperature with differences smaller than 1 °C during most of the days, with the exception of plots D-P1 and D-P3 (Fig. 5). There were no significant differences in the mean temperature among plot A-Polli9 with a high number of living snails (15–16 individuals), plot A-Polli3 with few living snails (one to two individuals), and other plots without living snails. The plots with a higher mean humidity (85%–100%) at the southern part of the hill did not harbour P. elephas as compared to plots with lower humidity (75%–93%) at the northern part of the hill (Fig. 6).

Association between the abundance of P. elephas and vegetation

Sixty-three taxa of vascular plants were recorded in the 17 plots. Species identifications were obtained for 46 species. The identity of 14 species could only be confirmed at the genus level, and the remaining three species could not be identified. Altogether 43 vascular plant species were recorded in only one plot, of which 27 species were singletons (Table 3). The number of species per plot ranged between three and 11 species, and the number of individuals ranged between four and 42 (File S4).

The abundance of four vascular plant species were positively correlated with the abundance of the land snails based on the null-hypothesis significance testing (p < 0.05) and Bayes factor (BF10) (Table 4, Table S7). Of these four species, Diospyros toposia var. toposoides (Ebenaceae) was the only plant species found in plots with and without living snails. Two plant species, namely, Croton cascarilloides (Euphorbiaceae) and Kibatalia laurifolia (Apocynaceae), were recorded only in two plots (A-P0 and A-Polli9). Another plant species, Mallotus peltatus (Euphorbiaceae), was recorded in only three plots (A-P0, A-P2 and A-Polli2). The total number of vascular plants and plant species was not correlated with the abundance of snails in the plots (Table 2).

Discussion

Synecology studies have focused on the association of habitat features and the composition of communities of land snails (Müller, Strätz & Hothorn, 2005). However, autecology studies on single species in their natural habitat are scarce. The two different approaches of ecology have developed independently, although the knowledge of both is necessary to understand the ecology of an individual population within a species or the whole ecosystem. A broader understanding of a species’ biogeography starts with the knowledge of the species’ autecology on a local scale (Hugall et al., 2002). Unfortunately, studies on the responses to environmental variables by individual species of large land snails in the tropical ecosystem are lacking (Horsák et al., 2007).

Spatial distribution and population density of P. elephas

The capture-mark-recapture technique has been used for estimation of population size and density for land snails (Blinn, 1963; Hänsel, Walther & Plachter, 1999; Standish, Bennett & Stringer, 2002; Parkyn, Brooks & Newell, 2014). Previous studies on other land snails showed that the capture rate using the CMR technique can be very high, with up to an 85% recapture rate after one year (Kleewein, 1999). In our study, the recapture rates varied by plot. Using this technique, we found that P. elephas can achieve high population densities, between three and four individuals per square meter, in suitable habitats at the northern part of the examined limestone hill. We could not find other population density studies on similarly-sized caenogastropod land snails for comparison. However, studies on similarly-sized pulmonate land snails showed that snails can occur in very low densities: less than one individual per m2 to very high densities of up to 100 individuals per m2 for Limicolaria martensiana (Owen, 1969). However, Owen (1969) did not investigate environmental factors that could determine the variation of densities in different population.

We could not find P. elephas in the southern part of the limestone hill, while large populations occurred on the northern part of the limestone hill. All examined plots were on the same limestone outcrop, with generally similar climatic condition, soil conditions, and land snail communities (Foon, Clements & Liew, 2017). Furthermore, there were no apparent differences in environmental variables in the microclimate, topography, habitat, and vegetation between the northern and southern part of the limestone hill.

Table 3 Checklist and the number of individuals per vascular plant species in each plot.

	Family and species	Plots	The total number of individuals of each species in 17 plots	The total number of plots where this species was present	
		Plots with livingPollicaria elephassnails	Plots without
livingPollicaria elephassnails			
		A-P0	A-P2	A-Polli2	A-Polli3	A-Polli7	A-Polli9	A-P14	B-P6	B-P7	C-P4	D11	D-P10	D-P12	D-P3	D-P8	D-P9	Intro-P1			
	Anacardiaceae																				
1	Mangifera sp. 1												1			1			2	2	
	Annonaceae																				
2	Annonaceae sp. 1													1					1	1	
3	Cananga odorata									1									1	1	
4	Orophea cuneiformis	2																	2	1	
5	Polyalthia sp. 1											4						2	6	2	
6	Polyalthia sp. 2		1																1	1	
7	Xylopia sp. 1							1								1			2	2	
	Apocynaceae																				
8	Kibatalia laurifoliaa	1					1												2	2	
	Bignoniaceae																				
9	Radermachera pinnata acuminata															1			1	1	
	Celastraceae																				
10	Euonymus javanicus						1												1	1	
	Clusiaceae																				
11	Garcinia cowa															1			1	1	
	Cycadaceae																				
12	Cycas clivicola					1													1	1	
	Dipterocarpaceae																				
13	Vatica kanthanensis					1							2					1	4	3	
	Ebenaceae																				
14	Diospyros frutescens											1							1	1	
15	Diospyros toposia var. toposoidesa	3		2	7	8	7					2		1				3	33	8	
16	Diospyros transitoria		2																2	1	
	Euphorbiaceae																				
17	Croton cascarilloidesa	2					1												3	2	
18	Macaranga gigantea														1				1	1	
19	Macaranga tanarius									1	3								4	2	
20	Mallotus barbatus							3											3	1	
21	Mallotus brevipetiolatus								7	12		3		5				2	29	5	
22	Mallotus peltatusa	6	7	2															15	3	
23	Pimelodendron griffithianum													1					1	1	
	Lamiaceae																				
24	Vitex siamica															2			2	1	
	Lauraceae																				
25	Dehaasia cuneata						1									1			2	2	
	Leguminosae																				
26	Archidendron jiringa					1													1	1	
27	Bauhinia sp.										1								1	1	
	Malvaceae																				
28	Hibiscus macrophyllus							8											8	1	
29	Leptonychia caudata									6		5						5	16	3	
30	Pterospermum acerifolium					1													1	1	
31	Sterculia rubiginosa													1					1	1	
	Melastomataceae																				
32	Memecylon lilacinum												8						8	1	
	Meliaceae																				
33	Aglaia grandis					1													1	1	
	Moraceae																				
34	Ficus fistulosa							1											1	1	
35	Ficus hispida										2								2	1	
36	Ficus schwarzii									1									1	1	
37	Ficus sp. 1										1								1	1	
38	Ficus sp. 2														2				2	1	
	Palmae																				
39	Arenga westerhoutii		4												1				5	2	
40	Borassodendron machadonis		1																1	1	
41	Caryota mitis							1											1	1	
	Pandanaceae																				
42	Pandanus piniformis												5			3	2		10	3	
	Phyllanthaceae																				
43	Bridelia tomentosa							1					10			25	8		44	4	
44	Cleistanthus gracilis			1	15				5				3						24	4	
45	Cleistanthus myrianthus		1															6	7	2	
	Primulaceae																				
46	Ardisia sp.											2							2	1	
47	Myrsine perakensis															1			1	1	
	Rubiaceae																				
48	Aidia densifolia			3	2								7			3	6		21	5	
49	Canthium sp. 1															3			3	1	
50	Canthium sp. 2																	4	4	1	
51	Saprosma sp.						1												1	1	
	Rutaceae																				
52	Micromelum minutum																	1	1	1	
53	Murraya paniculata				1	1	1			1									4	4	
54	Rutaceae sp. 1																2		2	1	
55	Rutaceae sp. 2																	1	1	1	
	Salicaceae																				
56	Homalium dasyanthum						2												2	1	
57	Homalium grandifllorum												1						1	1	
	Sapindaceae																				
58	Paranephelium spirei	1																	1	1	
	Sapotaceae																				
59	Isonandra perakensis												2						2	1	
	Violaceae																				
60	Rinorea bengalensis	2	2	3	4	3	2		5			2		7				6	36	10	
	Indet																				
61	Indet sp. 1						2												2	1	
62	Indet sp. 2													1					1	1	
63	Indet sp. 3													2					2	1	
Notes.

a Plant species were a positive correlation between the abundances of plants and snails was found.

Table 4 Correlation between the abundance of Pollicaria elephas for each plot with the abundance of the vascular plant species.

Family	Species	Occurrence in the 17 plots	Bayesian Kendall’s Tau Correlations	BF10	Kendall’s Tau Correlations	p-value	
Anacardiaceae	Mangifera sp. 1	2	−0.243	0.735	−0.243	0.295	
Annonaceae	Polyalthia sp. 1	2	−0.239	0.714	−0.239	0.296	
Annonaceae	Xylopia sp. 1	2	−0.243	0.735	−0.243	0.295	
Apocynaceae	Kibatalia laurifolia	2	0.527*	17.738	0.527*	0.023	
Dipterocarpaceae	Vatica kanthanensis	3	−0.050	0.320	−0.050	0.826	
Ebenaceae	Diospyros toposia var. toposoides	8	0.519*	15.604	0.519*	0.015	
Euphorbiaceae	Croton cascarilloides	2	0.499*	11.617	0.499*	0.030	
Euphorbiaceae	Macaranga tanarius	2	−0.239	0.714	−0.239	0.296	
Euphorbiaceae	Mallotus brevipetiolatus	5	−0.398	3.139	−0.398	0.070	
Euphorbiaceae	Mallotus peltatus	3	0.547*	23.867	0.547*	0.016	
Lauraceae	Dehaasia cuneata	2	0.162	0.454	0.162	0.485	
Malvaceae	Leptonychia caudata	3	−0.302	1.167	−0.302	0.187	
Palmae	Arenga westerhoutii	2	0.060	0.325	0.060	0.794	
Pandanaceae	Pandanus piniformis	3	−0.298	1.133	−0.298	0.187	
Phyllanthaceae	Bridelia tomentosa	4	−0.350	1.854	−0.350	0.116	
Phyllanthaceae	Cleistanthus gracilis	4	0.131	0.397	0.131	0.556	
Phyllanthaceae	Cleistanthus myrianthus	2	0.020	0.310	0.020	0.931	
Rubiaceae	Aidia densifolia	5	0.000	0.308	0.000	1.000	
Rutaceae	Murraya paniculata	4	0.370	2.279	0.370	0.112	
Violaceae	Rinorea bengalensis	10	0.257	0.809	0.257	0.223	
Notes.

* BF10 > 10.

** BF10 > 30.

*** BF10 > 100

* p < 0.05

** p < 0.01

*** p < 0.001

a The variance in Macaranga tanarius is equal to 0.

However, it is unclear why the snails from high-density spots did not migrate to the other spots at the same hill with similar habitat. The dispersal ability of this species was not determined in this study. Nevertheless, the dispersal distances for other similar-sized land snails are very short, ranging from meters to tens of meters per year (Baur, 1986; Schilthuizen et al., 2005; Edworthy et al., 2012; Ozgo & Bogucki, 2011; Kramarenko, 2014). One possible explanation could be the homing behaviour found in certain snail species (Rollo & Wellington, 1981; Tomiyama, 1992; Stringer, Parrish & Sherley, 2018) with highly specialised habitat requirements. These species are not migrating far from their favoured spot and can have narrow-ranged endemics occurring unevenly across a large landscape. Another possible explanation may be that an unfavourable habitat prevents dispersal of this species to isolated spots with suitable habitat.

Although the environmental variables included in this study were unlikely to determine the absence and presence of this species in different parts of the hill, the heterogeneity of population densities in the plots at the northern part of the hill showed that higher abundance of P. elephas could be associated with lower soil temperatures. This is expected as P. elephas is a ground-dwelling land snail. From our observation on the snails’ behaviour in the field and in captive populations, snails were active during the night where they were seen feeding on leaf litter. In the day, the snails could be found burying themselves underneath leaf litter. Living snails in the field were never found attached to vegetation or rocks above ground. Hence, we can assume that a constant and relatively low soil ground temperature and a lower air temperature and higher humidity at night is important for the population to thrive File S8).

Association between snail abundance and abundance of plant species

Previous studies on the association between plants and specific land snail species have been conducted outside of the tropical regions (Blinn, 1963; Pollard, 1975; Cowie, 1985; Hänsel, Walther & Plachter, 1999; Standish, Bennett & Stringer, 2002; Burrell, Scott & Yen, 2007; Horsák, Škodová & Cernohorsky, 2011; Parkyn & Newell, 2013; Parkyn, Brooks & Newell, 2014). Most of these studies suggested that plants act as shelters for land snails (Blinn, 1963; Pollard, 1975; Cowie, 1985; Standish, Bennett & Stringer, 2002; Burrell, Scott & Yen, 2007; Parkyn, Brooks & Newell, 2014).

The abundance of P. elephas is positively associated with a relatively common vascular plant, Diospyros toposia var. toposoides, on the limestone hill. The other three comparatively uncommon species, namely, Croton cascarilloides, Kibatalia laurifolia, and Mallotus peltatus were also positively correlated with the snails’ abundance. All of the three species are only found in plots with living snails (Table 3). However, it is important to note that due to the nature of high heterogeneity of vascular plants in the forest, as well as the 43 vascular plant species that were recorded in only one plot, there were only two plots for Kibatalia laurifolia and Croton cascarilloides, respectively, and three plots for Mallotus peltatus. Also, this indicates that the presence of the three plant species is not necessary for the presence of the snails as these plants were not found in other plots where snails were present. Hence, future studies of carefully-selected sites with different abundances of living snails and our identified species should be conducted to verify the possible causal relationship.

A plausible explanation for this relationship could be that the leaf litter from these plant species is suitable for the snails’ diet. However, it is also possible that rather than a direct causal relationship, both plants and snails prefer the same parameters of the local environment. A specially designed experiment is needed to test these hypotheses. To our knowledge, there were no in situ experiments on food preferences of land snails conducted in the field. However, there have been laboratory experiments conducted with decaying leaves of selected plant species (Puslednik, 2002; Proćków et al., 2013). In situ experiments on food preferences in a tropical rainforest are challenging because identifying leaf litter from plants is difficult, as plants are very species rich even within a small area, as was shown in this study (Crowther, 1982; Crowther, 1987a; Crowther, 1987b). We cannot rule out the hypothesis that plants and leaf litters provide shelter for land snails (e.g., Blinn, 1963; Pollard, 1975; Cowie, 1985; Standish, Bennett & Stringer, 2002; Burrell, Scott & Yen, 2007; Parkyn, Brooks & Newell, 2014).

There were single-species land snail studies to investigate the effects of vegetation on the population density in non-tropical (Hänsel, Walther & Plachter, 1999; Horsák, Škodová & Cernohorsky, 2011; Barrientos, 2000; Barrientos, 2019; Caldwell et al., 2014) and tropical regions (Barrientos, 2000). However, these studies relate the abundance of land snails, either with the general characteristic of vegetation structures (e.g., herbaceous vegetation in Hänsel, Walther & Plachter (1999); sparse herb vegetation in Horsák, Škodová & Cernohorsky (2011); thickness of herbaceous vegetation in Barrientos (2000)) or common plant species in the study sites (e.g., Barrientos, 2000; Caldwell et al., 2014). These studies could not establish proof of causation between specific plant species and the abundance of land snails. Although we could not confirm the in-depth association between the vascular plant species and snail feeding ecology, we were able to identify candidate plants to be included in future experiments. It would be worthwhile to investigate the pH and nutrient content of the plant species as the possible food sources and shelters for the land snail since there was an association between snail abundance and particular plant species.

Other factors that may affect the distribution and density of land snails

The distribution and density of P. elephas may be influenced by factors that were not investigated in this study, such as calcium availability, pH of the substrates, and predators. In the non-limestone forests, calcium availability plays a major role in determining the population density of snails (Gardenfors, 1992; Graveland & Van der Wal, 1996; Skeldon et al., 2007). However, we assumed that calcium availability may not vary significantly because all the plots were located next to the limestone outcrops (Crowther, 1987a); therefore, calcium availability may not be a factor that requires further attention among the plots.

Typically, the pH of the substrates (soil and leaf litter) will be affected by the bedrock, and the variation of pH among the plots on the limestone hills were very small (e.g., Crowther, 1982). However, leaf pH and leaf litter can vary significantly among different plant species (Cornelissen et al., 2011; Tao et al., 2019). However, a study of other land snails in another part of the world suggested that snail density correlated with calcium content and, to a lesser extent, with the pH of the litter layer (Graveland & Van der Wal, 1996).

Lastly, predation pressure could also explain snails’ density distribution patterns (Abramsky et al., 1992; Meyer & Cowie, 2010; Gerlach et al., 2020). Unfortunately, data on population densities and life histories for predators were not available from this study.

Conclusion

We determined the ecological aspects of P. elephas in terms of the habitat, topography, microclimate, and vegetation variables. We also found that ground temperature and a few vascular plant species were positively associated with snail abundance. Although our study was limited by its short duration and the absence of replicate sites on other hills, our findings can be used to formulate testable hypotheses when another population of this snail is found on further sites. After this exploratory study, we suggest a more focused, hypothesis-driven study to determine: (1) how the microclimates variations affect the land snails’ activities during the day and night; (2) the roles of the four vascular plant species that were found associated with living snails as food or shelter.

Supplemental Information

Supplemental Information 1 Raw data of pilot study and capture-mark-recapture study in the plots

Click here for additional data file.

Supplemental Information 2 An R script for PCA analysis and plotting chart of microclimate data

An R script for principal component analysis (PCA) to assess the degree of habitat heterogeneity among the 17 plots based on the two habitat variables (leaf litter thickness, and canopy cover), four topographic variables (elevation, slope, aspect, and ruggedness index) and two vegetation variables (number of vascular plant individuals and number of vascular plant species). The dataset of these variables can be found in File S4. In addition, the R script for plotting the microclimatic data to examine the variability of the climate data in File S6.

Click here for additional data file.

Supplemental Information 3 The dataset and the output of the analysis in JASP format for correction tests between the abundance of snails and each of the habitat, vegetation, and topography variables

The dataset and the output of the analysis can be viewed by using JASP software version 0.12.2 (JASP Team, 2020).

Click here for additional data file.

Supplemental Information 4 The dataset for the habitat, topographic, vegetation variables

This dataset can be analysed by using an R script of File S2 for principal component analysis (PCA) to assess the degree of habitat heterogeneity among the 17 plots based on the two habitat variables (leaf litter thickness, and canopy cover), four topographic variables (elevation, slope, aspect, and ruggedness index) and two vegetation variables (number of vascular plant individuals and number of vascular plant species).

Click here for additional data file.

Supplemental Information 5 Principal components analysis (PCA) plots of the third axis with the first and the second axis, respectively, for habitat, topography and vegetation variables for all of the 17 plots in a limestone hill

The colour of the plots’ label represents the plot location on the centre, northern and southern parts of the limestone hill. Living Pollicaria elephas land snails were not found in any of the plots in the southern and central part of the limestone hills. In contrast, the snails were found in the six of the seven plots on the northern part of the same limestone hill (except A-P14 plot).

Click here for additional data file.

Supplemental Information 6 Microclimate dataset of the plots

This dataset consists of three datasets, namely, soil temperature, ambient temperature and ambient humidity of the plots. The R script in File S2 can be used to reproduce the Figs. 4, 5 and 6.

Click here for additional data file.

Supplemental Information 7 Correlation between the abundance of land snails Pollicaria elephas for each plot with the abundance of the associated four vascular plant species

Click here for additional data file.

Supplemental Information 8 Soil temperature, air temperature and air humidity variations for every two hours of all the days in July 2018 in the plots

A figure and raw data for mean soil temperature, air temperature and air humidity for every two hours of all the days in July 2018 in the plots.

Click here for additional data file.

We thank Jaap Vermeulen and the two anonymous reviewers for their constructive comments. We want to thank Richard Chung Cheng Kong, the curator of The Herbarium in Forest Research Institute Malaysia (KEP), for allowing us to access the herbarium and its facilities. Postar Miun, a field botanist from Forest Research Center (FRC), has helped tremendously identify plant voucher specimens, for which we are grateful.

Additional Information and Declarations

Competing Interests

Author Contributions

Data Availability

The authors declare there are no competing interests. Mohamad Afandi Mat Said is employed by Associated Pan Malaysia Cement.

Thor-Seng Liew and Chee-Chean Phung conceived and designed the experiments, performed the experiments, analyzed the data, prepared figures and/or tables, authored or reviewed drafts of the paper, and approved the final draft.

Mohamad Afandi Mat Said conceived and designed the experiments, performed the experiments, authored or reviewed drafts of the paper, and approved the final draft.

Pui Kiat Hoo performed the experiments, authored or reviewed drafts of the paper, and approved the final draft.

The following information was supplied regarding data availability:

The raw data is available in the Supplemental Files.

The dataset and the output of the analysis (File S3) can be viewed by using JASP software version 0.12.2 (JASP Team, 2020).

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
