# Peer review of "Distribution and abundance of the land snail Pollicaria elephas (Gastropoda: Pupinidae) in limestone habitats in Perak, Malaysia"

_PeerJ, doi:10.7717/peerj.11886_

## Round 0.1 · original submission · Major Revisions

The reviewers highlight the strengths of your submitted manuscript and have proposed a number of changes to improve the quality. Please consider the proposed changes as mandatory for final acceptance of your manuscript.

Reviewer 1 ·

Basic reporting

English is clear, and generally, the paper is clearly written.

Literature: I think the literature survey is not sufficient. The following sentence is simply not true: "Unfortunately, studies of individual species responses to its environmental variables are still lacking (Horsák et al., 2007)".
For example, species of the genus Vertigo have been studied in Europe with every imaginable methods, mostly in Poland and England and Scandinavia. The same is true for some clausiliid species such as Bulgarica cana and Ruthenica filograna, and Albinaria is the southern Balkans. I strongly advise the authors to perform a more detailed literature survey, because ecological studies focusing on single species are all over.
A land snail ecological paper without any citations of studies about Vertigo? Looks very strange.

Experimental design

Experimental design: Looks superficially good, and indeed provides good insights, but some parts are not described in sufficient details. For example, what about the size (age) of snails you collected? How many of them were juveniles? In this snail species the newborn babies are just a few mm large. How small were the smallest marked specimens?

Validity of the findings

The main result that the snail lives in the northern part of the hill but not on the southern part, and its explanation by limited dispersal ability sounds interesting, but I simply don't believe it. As a field person I known that collecting on the northern part of limestone hills will always provide more specimens and species than on the southern part, even if I cannot support this with published evidence. Literature survey, however, will surely provide some papers that state this. It is usually said that strong statements (like yours about the fine-scale distribution of P. elephas, and its explanation) need strong evidence.

The microchabitat survey was done only in a single month. What about the habitat characteristics during the entire year? The distribution of a species is determined by habitat characteristics throughout the whole year. I am not saying that your results are not publishable, just your conclusions should be smoothened since your field data are rather limited.

Additional comments

General comments: More thorough literature survey, and smoothen your conclusions please. Otherwise your data provide nice addition to our knowledge on habitat preferences of terrestrial snails.

Reviewer 2 ·

Basic reporting

The English is partly poor and terms used are not always clear.
(The authors will find some corrections in the pdf)

Experimental design

The research questions should be better defined.
(The authors should provide more background information on the study area, the taxon and on patchy distribution patterns of species in limestone habitats)

The methods are partly not described with sufficient detail and are partly difficult to follow.
(The authors will find specific comments on that in the 'general comments' and the pdf)

Validity of the findings

Some statements regarding the results of this study are very bold and partly too speculative. Limitations of this study are not discussed thoroughly.
(The authors will find specific comments on that in the 'general comments' and the pdf)

Additional comments

This study provides information on the small-scale distribution of a land snail species inhabiting a limestone hill in Perak, Malaysia, and gives valuable insights on the abundances, population sizes and distribution patterns as well as the association with environmental parameters. In the following you will find some more general comments on the study. Attached you will find a pdf, which includes more specific comments and corrections on the manuscript text as well as on some tables and figures.

The title (‘Some ecological aspects…’) is very unspecific. Consider changing it to fit it more to your specific results, e.g. by using ‘distribution and abundance’.

The introduction should me more focused on the aim of the study. E.g. the part about general species conservation in limestone habitats is too unspecific for this study and is not focus of the discussion later. In addition, this study focusses on patchy distribution patterns of species within an area. A section about this topic would strengthen the introduction and discussion.

A justification of the study area should be made. You studied only one limestone hill, but I guess this makes sense for the aims of your study. Please make this clearer (please see my comment in manuscript).

It would be helpful to say a few words about why you studied this species of land snails. For example, it is not known from the introduction that this species is ground-dwelling.

Your study consists of two sampling session events; the ‘pilot survey’ and the ‘CMR sessions’. It is quite difficult to follow, which plots were selected for each survey and where and when the environmental variables were recorded (see also my comments in the manuscript). It would be helpful to make this clearer. You could also think about giving a table with all study sites, showing which survey was done and when and which variables where recorded and when.

Also, I would recommend to rename the ‘pilot survey’, as this is not a preliminary study (as far as I understood), but a full survey of equivalent importance to the subsequent survey. You could think of a name which is based on the data you collected, such as ‘survey of presence/absence data’.

A statement about which survey was done to tackle which research question, would be very helpful.

More details about the marking of snails in the CMR session would be helpful (see comment). Also, it is not clear which snails were marked in Additional file 1 (see comment).

The topographic features, recorded in this study (e.g. aspect and ruggedness index) should be explained.

Terms are used in an unclear way throughout the manuscript. E.g. ‘population ecology’, ‘vegetation composition’. (please see my comments in manuscript). These terms should be explained and used in a comprehensible way.

Tables should include information about the respective units of the values given. For example, they are missing in table Additional File 4 (see comment in manuscript).

You state that soil temperature is explaining the occurrence of the studied snail. Although you tested other parameters for a correlation which snail abundance, you did not show a test for this parameter. I find this statement a bit speculative and you should be more careful with this interpretation. In addition, it might help to test for a correlation (same for air temperature and humidity).

In addition to my comment above, you could show a graph of the correlation of the plants that were identified as being correlated with the abundance of the snails.

Some statements in the discussion are relatively bold (e.g. that ‘all Malaysian land snails have been little studied…’) and should be made more careful. It would help here to focus more on the own study and on ecology, population densities, small-scale distributions or habitats.

Some statements about the results of this study are also very bold and partly speculative (see comments in the manuscript). For example, you say in the conclusions that the habitat, topography, microclimate, and vegetation variables affect the occurrence of the studied species. What you found, is a correlation with some of the variables. But for most you did not find any correlation. Try to make statements more careful here and be more specific about what your data show.

In another example you state that the leaf litter from some plants could be suitable food sources for the snails. Is there any evidence for that? You mentioned some (unpublished?) observations that the snails feed on leaf litter. Is this statement derived from these observations? Please make this part clearer.

I miss in the discussion a part about the statistics and the limitations of this study. E.g., you have a very high number of plant species in your plots. And species are differing a lot in different plots. Could the correlation with the abundance of snails you found for some plants just be caused by random effects?

Include a part about other variables you did not measure, but which are know to be important for land snails (e.g. pH or calcium availability). Is there a reason why you did not measure them? What about the kind of rock? Foon et al 2017 mentions that there are differences in the rock habitat in this area. You should discuss them. Are you sure that the environmental variables you recorded are important for the occurrence of your species or did you miss some parameters, which could also explain the absence of this species in some of the plots?

The statements in the conclusion are quite bold on one hand, and on the other they are repetitions of what was said earlier. You could strengthen the conclusion by including a part on species conservation. E.g. what do your results add to the knowledge about species (or population) conservation in limestone habitats? How useful is it to do species inventories based on only very few localities? What do your results specifically mean for conservation of the studied species; is it enough for this species to protect suitable habitats?

The English is relatively poor. I did some corrections in the text but it would be good if a native speaker would check the final text again.

Annotated reviews are not available for download in order to protect the identity of reviewers who chose to remain anonymous.

·

Basic reporting

Advice: Publish after comments under C) and D) (see below) have been addressed by the authors, and the comments under A), B) and E) have been considered by the authors.

Experimental design

A) Suggestion to reorganize the paper a little. Starting at line 75 the initial observation is outlined (patchy occurrence of Pollicaria, and its absence on a part of the limestone hill). I suggest that the research question is spelled out after line 79, and that this paragraph (starting at line 75) is included in a separate chapter. The paragraph starting at line 80 could be added as introduction to the chapter Material and methods.

Considering the somewhat inconclusive results, it may be useful to write in terms of the ‘initial hypothesis to be tested’ in this paper (‘Environmental parameters determine the patchy distribution of Pollicaria’) to which the answer is, in the chapter conclusions ‘well, eh, none that we measured, except perhaps soil temperature’. Then the authors could speculate about what else stops the animals from crowding the hill, such as homing behavior.

Validity of the findings

The paper is based on rigorous field work to explain an initial observation. The resulting dataset seems, unfortunately, to yield somewhat inconclusive results, as happens sometimes. This outcome is perfectly acceptable to the Journal, as stated under ‘validity of the findings’.

Additional comments

B) Title seems to general, too vague; it does not cover the contents of the paper. A suggestion, using a sentence in the summary: ‘Effect of environmental variables on the abundance of the land snail Pollicaria elephas (Gastropoda: Pupinidae) in a limestone habitat in Malaysia’. Or, even more in line with the initial hypothesis to be tested: ‘Do environmental parameters determine the patchy distribution of Pollicaria (Gastropoda: Pupinidae) in a limestone habitat in Malaysia?’

C) The text is often verbose. I suggest going through the text sentence by sentence and take out redundant words and rephrase sentences. Examples:
161: ‘before proceeding to statistical analysis’: ‘proceeding to’ can be deleted
177: ‘There were no living snails nor empty shells found in all 11 plots at the southern and the central 178 part of the limestone hill. On the other hand, we recorded living snails and empty shells in six of 179 the seven plots at the northern part of the hill’ (50 words): ‘Individuals were found in six out of seven plots on the northern part of the hill; none were found in plots on the central and southern part’ (27 words). The information that individuals include living animals as well as empty shells is given earlier.
227: ‘There are 43 species of vascular plant that…’ . ‘Altogether 43 vascular plant species… ’

D) Language needs to be checked for incorrect grammar and syntax.

E) There may be some glitches in logic. The manuscript should be checked throughout. Possible examples (Please note that I am a geologist and taxonomist, not an ecologist, and read this text as an outsider):
46: ‘Of the 64 critically endangered terrestrial animal species in Malaysia listed in IUCN Red List, 26 belong to land snail species’. Could it be that the high number of snails on the list is because other group of limestone invertebrates are less systematically studied? The suggestion here seems that limestone land snails are most affected, while the message the authors may have in mind is that limestone biodiversity is threatened, altogether thousands of species (including the tiniest arthropods etc.), and among these many narrow endemics.
200: Soil temperature: Measurements were done in July, the northern summer during which the North flank of the steep hill receives more sunlight. Could it be that in December the South flank generally receives more sunlight, and that measurements during that period would yield opposite results?
Fig. 4 seems not very conclusive to me as far as the relation between soil temp and # individuals is concerned.
276: ‘there were no obvious differences in environmental variables in terms of climate, topography, habitat, and vegetation between the northern and southern part’: but authors claim that soil temperature is generally higher on the northern flank.
284: Stringer et al study homing behavior of Placostylus, different mollusks. The most parsimonious explanation may be that the animals find it difficult to bridge areas with less favorable habitat to isolated spots with preferred habitat.
304 to 310: A food relation between abundance of Pollicaria and the plant species may be a step too far. The more parsimonious reasoning may be that both plants and snails prefer the same aspect of the local environment (an aspect not covered in this study, perhaps).
336: Fig. 4 seems to suggest, vaguely, the opposite

---

## Round 0.2 · Major Revisions

Although both reviewers confirm improvements in the revised manuscript, there are still a number of linguistic and content-related points that require intensive revision. I therefore ask you again to revise your manuscript, taking into account the specific advice given by the reviewers.

Reviewer 2 ·

Basic reporting

The English has improved, but in many cases (especially in the newly added text parts) the language is still not clear and needs further work. Most of my General Comments (see below) refer to language issues. Again, it would really help if a native speaker could check the text, especially any newly added text parts.
In addition, there are still many inconsistencies in the terms and names used.

Experimental design

The research question is much better defined now. The methods are clearer and more detailed, however, there are still some missing and ambiguous parts that need further improvement (see my General Comments).

Validity of the findings

I do think the results are worth to publish, but the number of plots, where snails were present, is low and the correlations found, are sometimes weak. Most statements concerning the correlations of variables with snail occurrences or abundances are thus far too bold and need to be scaled down considerably (see my General Comments for specific examples). In addition, all limitations need to be addressed very clearly (the changes on the manuscript that have been done are still not sufficient). The comparison of other studies on land snails with own data should be more specific and discussed in a meaningful way.

Additional comments

Please see the attached PDF file for the general comments.

Annotated reviews are not available for download in order to protect the identity of reviewers who chose to remain anonymous.

·

Basic reporting

The English is, if not always strictly following grammar, clearly understandable. I suggest a number of small improvements in the text, and I advise a check by a native speaker to put the linguistic details right.
Generally, the reasoning in the manuscript is unambiguous; I left comments where I think that the text can be improved.
The structure of the manuscript follows professional standards. Figures and tables are of good quality and relevant to the text. The paper is self-contained.

Experimental design

The paper contains original research relevant to the Journal. The authors identify a gap in our knowledge, describe experiments to fill that gap and discuss the results, following prevailing standards and in sufficient detail for replication experiments.

Validity of the findings

Underlying data are provided. I cannot judge the statistical aspects of the experiment. Conclusions and speculations are separated.

Additional comments

I left comments and some improvements of the text with track changes in the manuscript, see file attached (attached to a separate mail, because the website refuses to accept the file as attachment). I think the paper can be accepted after the issues in the comments have been addressed, and after a language check-up of the text.

---

## Round 0.3 · Minor Revisions

We are almost there. The reviewer suggests a few minor corrections that you should take into account.

Reviewer 2 ·

Basic reporting

The English has clearly improved and the language is clear and unambiguous now.

Experimental design

The experimental design and all methods are described with sufficient detail now.

Validity of the findings

The results are well described and discussed. The limitations are addressed clearly now.

Additional comments

Please see attached document for my comments/corrections.

Annotated reviews are not available for download in order to protect the identity of reviewers who chose to remain anonymous.

---

## Round 0.4 · accepted · Accept

Thank you very much for the thorough revision of the manuscript and for the acceptance of the reviewer's comments.